# COVID-19 Pandemic: Perspective of an Italian Tertiary Care Pediatric Center

**DOI:** 10.3390/healthcare8030311

**Published:** 2020-09-01

**Authors:** Daniele Donà, Carlo Giaquinto, Eugenio Baraldi, Alessandra Biffi, Piergiorgio Gamba, Anna Maria Saieva, Luca Antoniello, Paola Costenaro, Susanna Masiero, Laura Sainati, Liviana Da Dalt, Giorgio Perilongo

**Affiliations:** 1Division of Pediatric Infectious Diseases, Department of Women’s and Children’s Health, University of Padua, Via Giustiniani 3, 35128 Padua, Italy; carlo.giaquinto@unipd.it (C.G.); paola.costenaro@phd.unipd.it (P.C.); 2Neonatal Intensive Care Unit, Department of Women’s and Children’s Health, University of Padua, Via Giustiniani 3, 35128 Padua, Italy; eugenio.baraldi@unipd.it; 3Pediatric Hematology, Oncology and Stem Cell Transplant, Department of Women’s and Children’s Health, University of Padua, Via Giustiniani 3, 35128 Padua, Italy; alessandra.biffi@unipd.it (A.B.); laura.sainati@unipd.it (L.S.); 4Pediatric Surgery Unit, Department of Women’s and Children’s Health, University of Padua, Via Giustiniani 3, 35128 Padua, Italy; piergiorgio.gamba@unipd.it (P.G.); lucamaria.antoniello@aopd.veneto.it (L.A.); 5Medical Staff Affairs, University Hospital of Padua, Padua, Via Giustiniani 3, 35128 Padua, Italy; annamaria.saieva@aopd.veneto.it; 6Pediatric Emergency Department Department of Women’s and Children’s Health, University of Padua, Via Giustiniani 3, 35128 Padua, Italy; susanna.masiero_01@aopd.veneto.it (S.M.); liviana.dadalt@unipd.it (L.D.D.); 7Department of Women’s and Children’s Health, University Hospital of Padua, Padua, Via Giustiniani 3, 35128 Padua, Italy; giorgio.perilongo@unipd.it

**Keywords:** COVID-19, children, pediatric department, hospital, infection prevention and control

## Abstract

Since February 2020, Italy has been faced with the dramatic spread of novel Coronavirus SARS-CoV-2. This impetuous pandemic infection forced many hospitals to reorganize their healthcare systems. Predicting a rapid spread of the SARS-CoV-2 virus within our region, the Department for Women’s and Children’s Health promptly decided (i) to revise the distribution of the clinical areas in order to create both designated COVID-19 and COVID-19-free areas with their own access, (ii) to reinforce infection prevention control (IPC) measures for all healthcare workers and administrative staff and (iii) to adopt the new “double-gate approach”: a phone call pre-triage and nasopharyngeal swab for SARS-CoV-2 detection before the admission of all patients and caregivers. Between 21 February 2020 till 04 May 2020, only seven physicians, two nurses and two of the administrative staff resulted positive, all during the first week of March. No other cases of intra-department infection were documented among the healthcare workers since all the preventive procedures described above were implemented. It is predicted that similar situations can happen again in the future, and thus, it is necessary to be more prepared to deal with them than we were at the beginning of this COVID-19 pandemic.

## 1. Introduction

On 21 February 2020, the first COVID-19 positive case in a 76-year old man in Vo, a small town in the province of Padua, Italy, was reported. The same evening, his two grandchildren were evaluated in the Pediatric Emergency Room of Padua University Hospital due to household exposure and that represented the first contact with the SARS-CoV-2 virus in our hospital.

The Department for Women’s and Children’s Health (W and CHD) of Padua is one of the eight departments according to which the hospital is articulated; functionally it is a “Children’s Hospital within a Hospital”, functioning as a tertiary pediatric care center. It is composed of 16 Divisions serving the Veneto Region, an area of 4,700,000 inhabitants of which the 13.5% represent the cohort of children aged 0–14 years. It counts 211 beds of which 104 (49%) are dedicated to the different pediatric units (of which 8 are for Pediatric Intensive Care), 75 (36%) to the neonatal areas (of which 31 are for Neonatal Intensive Care) and 32 (15%) to the various pediatric surgical divisions (Figure 1). The staff of the Department is composed of 222 physicians (including 100 residents), 330 nurses and 75 other personnel. In 2019 the hospital admissions had been 10,428, the outpatient visits 180,000, births 2820 and accesses to the Pediatric Emergency Department (PED) slightly over 24,000. Considering that every child entering the W and CHD is accompanied by at least 1 caregiver, the number of accesses should actually be doubled. The clinical wards for the inpatient activity and ambulatory services are hosted in a five story high main building except for the Onco-Hematology and Stem Cell Transplant Unit, the Surgical Divisions and the Neonatal Units which are operating in satellite buildings attached to the main one (Figure 1).

Predicting a rapid spread of the SARS-CoV-2 virus within our region, in the afternoon of February 24th the Chief Executive Officer (CEO) of Padua University Hospital called for an emergency meeting with all the department chairmen and the mandates received were:to ensure the protection of the healthcare workers, as the top priority;to rigorously implement all the conventional rules emanated by the WHO for preventing the infectionto minimize the risk of admitting into hospital asymptomatic COVID-19 positive patients;to adapt/transform some hospital areas in order to be able to admit and treat suspected/confirmed; COVID-19 pediatric patientsTo ensure continuity of treatment when appropriate and needed.

This impetuous pandemic infection put a tremendous amount of stress on many hospitals which were forced to reorganize their healthcare systems [1]. In just 10 days, the Chinese government built two hospitals in Wuhan for treating confirmed COVID-19 cases. As time passes, China and other countries completely reorganized their healthcare systems by rapidly building new temporary hospitals [2,3]. In our setting, the most rapid and efficient strategy was to completely reorganize pre-existing hospital areas in order to face with the emergency.

Therefore, to fulfill all the given tasks, the W and CHD promptly decided:-To revise the distribution of the clinical areas in order to create both designated COVID-19 and COVID-19-free areas with their own access.-To reinforce infection prevention control (IPC) measures for all healthcare workers and administrative staff-To reinforce IPC measures for patients with a new “double-gate approach”: a phone call pre-triage and nasopharyngeal swab (NPS) for SARS-CoV-2 detection before the admission of all patients and caregivers.

One of the key elements, among the many, which composed the comprehensive overall policy the University Hospital of Padua adopted to contain the COVID-19 epidemic, was to free the request of diagnostic NPSs on a 24/7 basis. Importantly, the routes to request NPS were highly facilitated and a very fast track (within three hours) was also made available to handle critical clinical situations.

Despite the fact that it was predicted that children could have been less frequently and less severely affected than adults [4], it also highly affected tertiary care pediatric institutions such as the W and CHD.

At the end of two months of the so called ”Phase I” of the COVID-19 pandemic that in Italy lasted between the last week of February till 4 May 2020, the W and CHD decided to run a preliminary critical assessment of the actions undertaken in this phase as per mandate of the CEO of the Hospital.

## 2. Multilevel Interventions to Prevent SARS-CoV-2 In-Hospital Spread

### 2.1. Redistribution of Clinical Areas in Order to Create COVID-19 and COVID-19 Free Areas with Their Own Access

To monitor all the people entering the W and CHD, the access to the building was modified in order to have only one single entering point to the Department for each building (without considering the PED entrance, which always remained available) (Figure 1). Moreover, right outside of each entry point an NPS station was set up to perform NPSs to all healthcare workers and patients with caregivers who were attended for urgent or non-deferrable hospitalizations.

The PED and the Pediatric Emergency ward were designated for the evaluation and treatment of suspected/confirmed COVID-19 cases. A pre-triage area was created at the immediate entrance of the pediatric emergency room, for all non-critical pediatric patients. The pre-triage area served to screen patients with epidemiological risk or clinical signs and/or symptoms of possible COVID-19 infection. This served to point them towards an ad-hoc separate entry pathway leading to a so-called COVID-19 area, totally separated from the rest of the emergency room with a good ventilation, where they could be visited. Furthermore, another 24/7 emergency area with two short-stay isolation rooms, was set up in the Onco-Hematology building for in-treatment oncologic and transplanted patients.

Our Pediatric Emergency Unit was transformed in an exclusive COVID-19 ward. The existing six two- or three-bed rooms, were all transformed into single-bed rooms. All patients in the COVID-19 ward were assisted by dedicated staff. The patient’s caregiver had to wear a surgical mask and was not allowed to exit the room. No visits were allowed.

Medical shifts were reorganized to allow a lower patient/staff ratio. Education of health care workers on preventive measures set up was guarantee, also leading to minimize occupational stress [5].

Residents’ rotations were suspended for two months and they continue to work in the same ward they were before the pandemic started. A dedicated COVID-19 team was created.

All conventional large face-to-face meetings were moved to telematic platforms and all administrative staff was directed to work remotely.

The presence of volunteers and play-therapists was interrupted. All school activities run in-house were also suspended. Gatherings of people close to all vending machines were banned.

### 2.2. Reinforcing of IPC Measures for All Healthcare Workers and Administrative Staff

All healthcare workers entering the Department hospital were required to have their body temperature checked on daily basis; the use of surgical masks was made mandatory, except for those working in the COVID-19 area who had to wear FFP2 masks and dedicated personal protective equipment at all times. Moreover, anesthesiologists had to wear FFP3 masks. A strict hand hygiene policy has been applied.

Starting from 4 March, a periodical screening with NPSs for all hospital personnel and administrative staff was implemented. Generally, NPS screening was performed on a 20-day basis, but for those healthcare workers dealing with COVID-19 patients, all anesthesiologists and those working with fragile patients (immunosuppressive children, children with chronic diseases or premature babies and/or the ones admitted to the neonatal intensive care unit) NPSs were performed every 10 days.

Initially, for those who had been in close contact with COVID-19 case, a 14-day home isolation was recommended. After the first two weeks of March, however, the rules changed and close contacts, if asymptomatic, were allowed to resume work upon obtaining an NPS every 48 h and then, from the beginning of April, every 5 days.

### 2.3. Reinforcing of IPC Measures for Patients with the New “Double-Gate Approach”: Phone Call Pre-Triage and NPSs for All Admitted Patients and Caregivers

On a daily basis, the head nurses were required to call in the families who were scheduled to be admitted to the hospital, within the 1–2 days prior to the admission to run a telephone pre-triage, based on a standardized questionnaire aiming to pick-up epidemiologic risk factors or signs and symptoms of COVID-19 infection in the patient or in his/her accompanying family member. In case of positive results, the admission was rescheduled.

Since SARS-CoV-2 infected patients could remain asymptomatic, after the telephone pre-triage, children and caregivers were requested to come to the hospital for receiving an NPS and then sent back home in order to be admitted as soon as the results were available (usually at the most within 48 h).

On the day of admission, all children and their caregivers were asked to self-complete the same questionnaire used for the telephone triage; they also had their body temperature checked. Inpatient access was allowed only upon obtainment of a negative test result for SARS-CoV-2 for both the patient and caregiver. Finally, all the people were encouraged to respect the social distances and a strict hand hygiene policy. It was requested to have only one caregiver for each patient and always the same for the entire duration of the hospitalization.

Differently, the child and the caregiver entering the hospital for an outpatient visit were screened only by the telephone pre-triage a couple of day before the visit. Upon entering the hospital, the same inpatients’ screening was applied. In order to guarantee the social distance in the waiting rooms, the seats were organized to respect at least one-meter of distance from each other and the appointments were distributed over a longer-than-usual time schedule with patient and caregiver’s place assignment.

In order to reduce overcrowding and ensure proper continuation of treatment, telemedicine and home-based treatments were increased to offer blood tests, antibiotic and chemotherapy, provided by doctors and nurses of the Onco-Hematology unit.

## 3. Results of Intervention

Since the end of February, the city of Padua, with a population of about 300,000 inhabitants, had 3844 confirmed infections with 281 SARS-Cov-2 attributable deaths [6].

In this ten-week period a total of 3382 NPSs were performed on healthcare workers. In total, 3371 (99.7%) of these came back negative. Only seven physicians (four consultants and three residents), two nurses and two of the administrative staff resulted positive, all during the first week of March (Table 1).

Upon reconstructing their recent histories of potential exposures, it turned out that ten of them (10/11, 91%) were unintentionally exposed to a SARS-CoV-2 infected person outside the hospital.

No other cases of intra-department infection were documented among the healthcare workers since all the preventive procedures described above were implemented. No doctor, nurse or other personnel working in the Department became infected; despite the fact that infected healthcare workers continue to attend the hospital for at least two days, before the positive results of the NPS became available, no other colleagues with whom they were working were infected.

Around six thousand pre-admission telephone pre-triages have been carried out, over 132 remote medical consultations and 275 accesses at the patients’ domicile have been performed.

Through the “double-gate” approach a total of 1885 pre-admission NPSs have been performed to children and their caregiver with non-deferable admissions (Table 2).

Among these, only three asymptomatic COVID-19 caregivers were identified. One was the mother of a child coming weekly to the day-hospital for an enzyme replacing therapy and the other two were the paucisintomatic parents of a newborn. These three people were also wearing a mask during their staying in the hospital. None of the personnel they got in contact with were infected, including their respective children. These three asymptomatic COVID-19 patients were then sent home to complete a quarantine and prevented to come into the hospital. Finally, none of the children who had an NPS done pre-admission turned out to be an asymptomatic positive carrier of SARS-CoV-2 virus.

In addition, it should be noted that thanks to the decision of postpone all elective admissions, to telephone triage and most likely also an element of fear of entering the hospital, the number of regular hospital admissions registered in the months of March and April 2019 in comparison to the ones registered in the same time year of 2020 dropped from 1306 to 937 and the day-hospital admission from 387 to 188.

Although the activity of the PED significantly reduced (−75%), from 21 February 1291 patients have been evaluated and 416 (32.2%) have been tested for SARS-CoV-2 virus. Three hundred (300/1291, 23.2%) were evaluated because of fever and/or respiratory or gastrointestinal symptoms. All patients received the COVID-19 test and for seven (7/300, 2.3%) this turned positive. All close contact with a confirmed COVID-19 case (24/1291) admitted to our PED were tested and four (4/24 16.7%) were found positive. Most children (66.8%) were referred for non-COVID-19 related problems and they were tested only in case of hospital admission. In total, 92/967 (9.5%) were tested, all with negative results.

## 4. Conclusions

The final result of this analysis clearly indicates that during the two months when the spread of the SARS-CoV-2 in Padua and in the Veneto region reached its peak and the numbers of people infected, at the time this analysis was ended, was still very high, the W and CHD remained a COVID-19-free environment. Indeed, in that time period no member of staff, resident, patient and her/his own caregiver during their in- or outpatient stay in the Department got infected by the SARS-CoV-2 infection.

It is difficult to pinpoint exactly what contributed to this success; we can only state that the comprehensive measures which were implemented resulted very effective at a time during which the entire city of Padua was locked and the “Entropy” of the Department decreased significantly. Furthermore, the expected small number of children infected by SARS-CoV-2 and thus the actual very small number of infected children who were admitted should also be considered in reading these positive results [7,8].

The main question which remains unanswered is indeed if the policy of freeing the use of NPSs and thus of using this test to screen all the people working and entering the Department played a major role in making the W and CHD a SARS-CoV-2-free Hospital. Relevant human and financial resources had to be invested. Two shifts of nurses, each one composed of three to four nurses, had to be fully dedicated to the NPS station in order to run it efficiently and smoothly. Other nurses, respectively working in the PED, in the Neonatology and in the Onco-hematology divisions were involved in obtaining NPSs to their own medical personnel considering that those units elected, for functional reasons, to deal with their doctors and nurses independently. On top of that, one should consider the financial cost of obtaining the NPSs and of the subsequent analyses.

Obviously, a direct cost-to cost confrontation between the one necessary for obtaining an NPS and the one necessary to deal with even a single doctor and/or nurse infected is unconceivable. However, the rationale for that policy, considered of course in the context of all the general and local procedures which were implemented, should be strongly re-read based on our findings.

It should be further acknowledged that, as already reported in the literature, these aggressive organizational and structural measures were generally well accepted [5]. The extensive use of PPE and the reorganization of separated areas with good ventilation were very effective in relieving the concern the staff of the Department and the patients and their caregivers experienced in those days.

This positive personal feeling was quite important in continuing to provide an effective care to the patients and in preventing the patients to not seek for medical care just for the fear of the SARS-CoV-2 infection.

It is predicted that similar situations can happen again in the future, and thus, it is necessary to be more prepared to deal with them than we were at the beginning of this COVID-19 pandemic.

All that is written above represents the content, aims and rationale of this publication.

## Figures and Tables

**Figure 1 healthcare-08-00311-f001:**
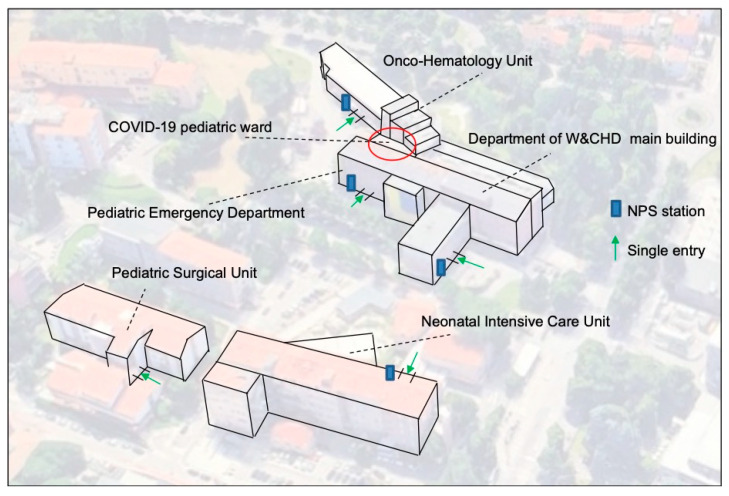
Distribution of clinical wards with their own single access and nasopharyngeal swab (NPS) station: a representative map.

**Table 1 healthcare-08-00311-t001:** Summary of NPS HCWs surveillance.

W and CHD Areas	Number of HCWs	Number of NPS	Number of Negative NPS	Number of Positive NPS
Main building	255	700	697	3
Onco-Hematology Unit	107	900	900	0
NICU	91	728	725	3
Pediatric Emergency Department	60	554	554	0
Pediatric Surgery Unit	80	500	495	5
Total	593	3382	3371	11

**Table 2 healthcare-08-00311-t002:** NPS Surveillance in Children and Their Caregivers for Non-Deferrable or Urgent Hospital Admissions.

W and CHD Areas	Number of Hospital Admissions (Including DH Activity)	Number of NPS (Patient + Caregiver)	Number of Negative NPS	Number of Positive NPS
Main building	377	720	719	1
Onco-Hematology Unit	95	188	188	0
NICU	93	426	423	2
Pediatric Emergency Department	92	185	185	0
Pediatric Surgery Unit	183	366	366	0
Total	840	1885	1882	3

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
