# Peer review of "COVID-19 Pandemic: Perspective of an Italian Tertiary Care Pediatric Center"

_healthcare, 2020, doi:10.3390/healthcare8030311_

Round 1
Reviewer 1 Report
This manuscript assessed the actions undertaken in a tertiary care pediatric institutions in Italy during the epidemic of COVID-19. The actions included multilevel interventions to prevent SARS-CoV-2 in-hospital spread which proved to be effective. I think this is an interesting and important topic, but there are some minor issues that need to be addressed.
- The Abstract section needs to be revi Currently it has too much information on background rather than measures, results and conclusions.
- Introduction: I would suggest the authors to the third - person narrative form, and introduce the hospital first.
- Closing remarks: Most of this section are about effects of the actions taken, probably divide it into results and conclusions.
- Line 163-164: there were 11 positive NPSs but this description did not add up to 11.
Author Response
This manuscript assessed the actions undertaken in a tertiary care pediatric institutions in Italy during the epidemic of COVID-19. The actions included multilevel interventions to prevent SARS-CoV-2 in-hospital spread which proved to be effective. I think this is an interesting and important topic, but there are some minor issues that need to be addressed.
- The Abstract section needs to be revi Currently it has too much information on background rather than measures, results and conclusions.
- Introduction: I would suggest the authors to the third - person narrative form, and introduce the hospital first.
- Closing remarks: Most of this section are about effects of the actions taken, probably divide it into results and conclusions.
- Line 163-164: there were 11 positive NPSs but this description did not add up to 11.
1. Authors’ answer:
Thank you for your comment, we have changed the abstract accordingly.
2. Authors’ answer:
Thank you for this comment. As suggested, the third-person narrative form has been used, in addition the hospital introduction has been moved early in the manuscript.
3. Authors’ answer:
Thank you for this comment. As suggested, we have split closing remarks in two separate sessions named “results of interventions” and “conclusion”.
4.Authors’ answer:
Thank you for your comment. We fixed it.
Reviewer 2 Report
This paper is timely during the COVID-19 pandemic. I have the following recommendations. I am happy to review again.
1) The authors seem to over-emphasize temperature measurement. COVID-19 patients are often afebrile. Please state the limitation of temperature measurement.
2) Is there regular COVID-19 testing of staff in the hospital? If yes, how often it can be done and how often it is done?
3) It is important to compare the hospitals built in other countries for COVID-19. China has rapidly built various hospitals. Can the authors discuss measures adopted by hospital in China whether this Italian hospital has not adopted. The following is a useful website:
https://www.archdaily.com/937579/a-closer-look-at-the-chinese-hospitals-built-to-control-the-covid-19-pandemic
4) For occupational health, the author needs to discuss other aspects including ventilation or concerns from management. The following study found these measures protect mental health of staff. Please discuss measures mentioned about this study and I hope this can be integrated in the Italian hospital:
Tan W, Hao F, McIntyre RS, et al. Is Returning to Work during the COVID-19 Pandemic Stressful? A Study on Immediate Mental Health Status and Psychoneuroimmunity Prevention Measures of Chinese Workforce [published online ahead of print, 2020 Apr 23]. Brain Behav Immun. 2020;S0889-1591(20)30603-6. doi:10.1016/j.bbi.2020.04.055
5) Does the hospital provide accommodation to staff to avoid infection to their relatives and family members?
Author Response
This paper is timely during the COVID-19 pandemic. I have the following recommendations. I am happy to review again.
1) The authors seem to over-emphasize temperature measurement. COVID-19 patients are often afebrile. Please state the limitation of temperature measurement.
2) Is there regular COVID-19 testing of staff in the hospital? If yes, how often it can be done and how often it is done?
3) It is important to compare the hospitals built in other countries for COVID-19. China has rapidly built various hospitals. Can the authors discuss measures adopted by hospital in China whether this Italian hospital has not adopted. The following is a useful website:
https://www.archdaily.com/937579/a-closer-look-at-the-chinese-hospitals-built-to-control-the-covid-19-pandemic
4) For occupational health, the author needs to discuss other aspects including ventilation or concerns from management. The following study found these measures protect mental health of staff. Please discuss measures mentioned about this study and I hope this can be integrated in the Italian hospital:
Tan W, Hao F, McIntyre RS, et al. Is Returning to Work during the COVID-19 Pandemic Stressful? A Study on Immediate Mental Health Status and Psychoneuroimmunity Prevention Measures of Chinese Workforce [published online ahead of print, 2020 Apr 23]. Brain Behav Immun. 2020;S0889-1591(20)30603-6. doi:10.1016/j.bbi.2020.04.055
5) Does the hospital provide accommodation to staff to avoid infection to their relatives and family members?
Authors’ answer:
Many thanks for this comment. We’ve followed the indications provided by the WHO. But of course, the role of the asymptomatic patients is extremely important. For this reason, we decided to emphasize this aspect at the begging of the paragraph in lines 283-284. We hope the reviewer agrees with us.
- Authors’ answer:
Covid-19 testing of hospital staff has been described in the paragraph, lines 266-271.
Authors’ answer:
As suggested, measures set up in China were mentioned in lines 117-121
Authors’ answer:
Thank you for raising this point. According to your suggestions, the text has been integrated – please refer to line 160-162 and lines 386-389
- Authors’ answer:
Unfortunately, our hospital didn’t provide any accommodation for medical staff, to avoid infection to their relatives.